# The mechanism of the plant roots' soil-reinforcement based on generalized equivalent confining pressure

Ping Guo[1,2,*], Zhenyao Xia[1,2,*], Qi Liu[3], Hai Xiao[1,2], Feng Gao[1], Lun Zhang[1], Mingyi Li[1,2], Yueshu Yang[1,2] and Wennian Xu[1,2]

[1] Key Laboratory of Geological Hazards on Three Gorges Reservoir Area (China Three Gorges University), Ministry of Education, Yichang, China

[2] Engineering Research Center of Eco-environment in Three Gorges Reservoir Region, Ministry of Education, China Three Gorges University, Yichang, China

[3] Key Laboratory of Mountain Hazards and Earth Surface Process, Institute of Mountain Hazards and Environment, CAS, Chengdu, China

[*] These authors contributed equally to this work.

## ABSTRACT

**Background**. To quantitatively evaluate the contribution of plant roots to soil shear strength, the generalized equivalent confining pressure (GECP), which is the difference in confining pressure between the reinforced and un-reinforced soil specimens at the same shear strength, was proposed and considered in terms of the function of plant roots in soil reinforcement.

**Methods**. In this paper, silt loam soil was selected as the test soil, and the roots of *Indigofera amblyantha* were chosen as the reinforcing material. Different drainage conditions (consolidation drained (CD), consolidation undrained (CU), and un-consolidated undrained (UU)) were used to analyse the influences of different root distribution patterns (horizontal root (HR), vertical root (VR), and complex root (CR)) and root contents (0.25%, 0.50%, and 0.75%) on the shear strength of soil-root composites.

**Results**. The cohesion ($c$) values of the soil-root composites varied under different drainage conditions and root contents, while the internal friction angle ($\varphi$ ) values remain basically stable under different drainage conditions. Under the same root content and drainage conditions, the shear strength indexes ranked in order of lower to higher were HR, VR and CR. The GECP of the soil-root composites with a 0.75% root content was 1.5–2.0 times that with a 0.50% root content and more than 5 times that with a 0.25% root content under the CD and CU conditions. The GECP in reinforced soil followed the sequence of CD $>$ CU $>$ UU. The GECP of the plant roots increased as confining pressure increased under CD and CU conditions while showed a complex change to the confining pressure under the UU condition.

**Conclusion**. It was concluded that the evaluation of plant root reinforcing soil based on GECP can be used to measure effectively the influences of roots on soil under different drainage conditions and root distribution patterns.

Corresponding author
Hai Xiao, oceanshaw@ctgu.edu.cn

## INTRODUCTION

Plant roots play an important role in improving the overall stability of the superficial slope soil and increasing the safety coefficient of the slope (*Zegeye et al., 2018*; *Zhou & Wang, 2019*). The plant root system is a complex and dynamic system, for which non-destructive monitoring is difficult, so it is always a challenging aspect to consider in research regarding the mechanism of plant root reinforcing soil.

At present, the evaluation of slope vegetation protection mainly includes mechanical and hydraulic mechanisms (*Gonzalez-Ollauri & Mickovski, 2017*; *Feng, Liu & Ng, 2020*). Based on both mechanisms, three vegetation protection theories were proposed, namely, mechanical reinforcement is provided by plant roots (*Jin et al., 2019*), the excess pore-water pressure in soil is dissipated by root water uptake (*Liu, Feng & Ng, 2016*) and soil matric suction is induced via plant transpiration (*Ng et al., 2013*; *Gadi et al., 2019*). The most obvious way in which vegetation enhances slope stability is root reinforcing.

The effect of root reinforcement on slope stability can be evaluated directly in terms of the additional shear strength provided by plant roots in reinforced soil. To analyse the effect of plant roots on slope stability, many in situ and laboratory tests have been carried out on vegetated soil (*Wu & Watson, 1998*; *Operstein & Frydman, 2000*), and corresponding analytical models for soil-root composites have also been developed (*Waldron, 1977*; *Waldron & Dakessian, 1981*; *Wu et al., 1988*). For example, a linear equation of root population density and soil shear strength was obtained (*Endo & Tsuruta, 1969*), in which the cohesion strength extending to the sliding layer has a stabilizing effect on shallow slopes, by in situ shear tests (*Gray & Ohashi, 1983*; *Greenway, 1987*).

In addition, some mechanistic models like the Wu-Waldron model, the modified Wu-Waldron model, the fiber bundle model, the root bundle model and have been developed to evaluate the additional shear strength provided by plant roots (*Wu, 1976*; *Waldron, 1977*; *Wu, McKinell & Swanston, 1979*; *Gray & Sotir, 1998*; *Pollen & Simon, 2005*; *Schwarz et al., 2010*). However, as the most classic and representative model, the Wu-Waldron model potentially significantly overestimates the actual cohesion of soil-root composites (*Waldron & Dakessian, 1981*; *Operstein & Frydman, 2000*; *Pollen & Simon, 2005*), because the Wu-Waldron model or the modified Wu-Waldron model is derived based on the assumption that plant roots are elastic and initially oriented perpendicular to the shear surface and that the friction angle of the soil is unaffected by the plant roots (*Waldron, 1977*; *Greenway, 1987*). Therefore, a correction factor ranged from 0.34 to 0.50 for roots of herbs and shrubs was proposed by *Schwarz et al. (2010)* to reduce the error of the Wu-Waldron model. The equation of generalized equivalent confining pressure (GECP) is derived based on the limit equilibrium state of reinforced soil and un-reinforced soil (*Huang et al., 2007*), in which the assumptions of root characteristics and root distribution can be ignored. The effect of root distribution or root shear failure angle on soil can be shown by the deviator of the failure principal stresses of reinforced and un-reinforced soil under the same confining pressure. Therefore, we try to introduce this method to assess the additional shear strength provided by plant roots.

Decisive factors controlling shallow landslides are the mechanical properties of the sloping soil characteristics (e.g., texture), frequency and duration of the rainfall, and plant species (root morphology) (*Matsushi, Hattanji & Matsukura, 2006*; *Normaniza, Faisal & Barakbah, 2008*). Rainfall may give rise to shallow landslides because it can increase in soil moisture content so that make the slope in instability stage when other conditions were the same. The effect of plant roots on the shear strength of vegetated soil significantly decreases because of the rainfall (*Normaniza & Barakbah, 2006*; *Jiang, Dong & Wang, 2009*).

Differences in depth, soil moisture content and root characteristics may result in a substantial change in soil shear strength. The effect of plant roots in reinforced soil is understood as an additional confining pressure to the soil, in excess of the traditional equivalent confining pressure. Therefore, the expression of generalized equivalent confining pressure (GECP) was derived to investigate the influence of root contents and root distribution patterns on the shear strength of reinforced soil under different drainage conditions (consolidation drained (CD), consolidation undrained (CU), and unconsolidated undrained (UU)) and was used to analyse the influences of different root distribution patterns (horizontal root (HR), vertical root (VR), and complex root (CR) patterns) and root contents in this research. This research provides new sight to assess the additional shear strength provided by plant roots for the soil-root composite.

# MATERIAL AND EXPERIMENTAL METHODS

## Experimental materials

In this paper, the soil was taken from the cutting slope of the first phase of the urban expressway along the Xiazhou Avenue in Yichang, China. The test soil was collected from 0.3 m below the surface, and the impurities in the soil were removed. The soil was air-dried, crushed and sieved through a 2.0 mm sieve. The soil had a silt loam texture with 24.08% sand (0.05–2.00 mm), 55.91% silt (0.002–0.05 mm), and 20.01% clay (<0.002 mm) contents, a 1.38 g $cm^{-3}$ bulk density, a 14.37% natural moisture content, a 2.78% air-dried soil moisture content, and a pH of 6.2.

*Indigofera amblyantha*, used widely in slope greening projects, were the roots selected as the reinforcing material. *Indigofera amblyantha* is a perennial deciduous shrub, and its growing period is approximately 6 months; it possesses strong drought resistance and barren resistance. These plants are the most common soil-water conservation plants in tropical and subtropical regions.

*Indigofera amblyantha* has a horizontally developed root system, including many branches and fibrous roots (Fig. 1), and the root diameter is mostly concentrated within 1.0–2.5 mm. In this paper, 50 plants of *Indigofera amblyantha* were excavated by the whole excavation method. Normal and straight roots were cut to lengths of 30 mm and 60 mm with scissors. The roots with an average diameter of 1.4–1.6 mm were chosen, of which the average tensile resistance and the average tensile strength were 62.10 N and 35.86 MPa, respectively.

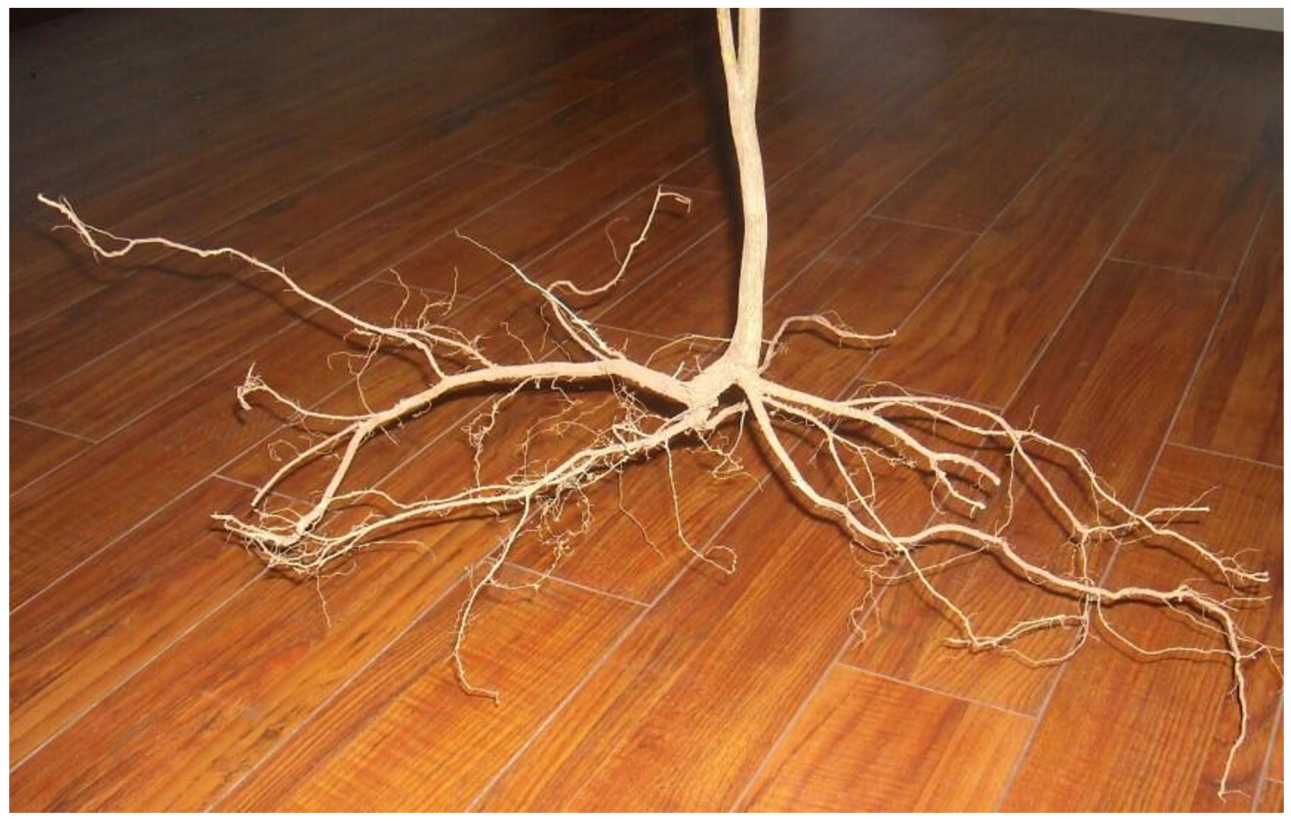

Figure 1 The root distribution of the *Indigofera amblyantha*.

## Experimental methods

The density and moisture content of the soil-root composites were set according to the actual situation of the test soil taken from the cutting slope (bulk density is 1.38 g $cm^{-3}$, and natural moisture content is 14.37%). The root contents (the ratio of the root mass to soil mass in the specimens) were set to 0.25%, 0.50% and 0.75%.

The *Indigofera amblyantha* has a horizontally developed root system, resulting most of the roots are in vertical stage on the slope. To evaluate the effect of root distribution patterns on the shear strength of reinforced soil and ensure the vertical roots are much more than the horizontal roots, the root distribution patterns were categorized into VR, HR and CR in this research (Figs. 2A, 2B, 2C). The first form (A) is VR with a root length of 60 mm; the second form (B) is HR with the root length is 30 mm; the third form (C) is CR with the ratio of the horizontal to vertical roots is 1:1 in mass (2:1 in number). In this paper, plant roots were organized in the centre of soil-root composites in three forms.

As most *Indigofera amblyantha* roots are concentrated within 0.5 m below Earth's surface. When the depth exceeds 0.5 m, the reinforcing effect of plant roots is not obvious because the root content is low (*Waldron & Dakessian, 1982*). Therefore, to effectively evaluate the GECP of plant roots in reinforced soil, three levels of confining pressure (50 kPa, 100 kPa and 150 kPa) are tested in this paper.

Soil-root composites were remoulded in a circular loading box of $\Phi$ 39.1 mm × 80 mm (Fig. 2D), which matched with the TSZ-1 strain-controlled triaxial compression apparatus. First, a suitable amount of test soil was weighed and placed in a container that could be sealed, and an appropriate amount of water was sprayed on the soil to reach the moisture content required in this work. Second, the test soil and water were fully mixed, and then the container was sealed for 24 h until the test soil was soaked completely. Third, the required amount of soil was taken from the sealed container and placed in the circular loading box mentioned above. Finally, plant roots were buried evenly in the soil, and the method of three-layer compaction was adopted to remould the soil-root composites in the circular loading box according to the standardized methods of soil mechanics test and specimen preparation. In addition, specimens of un-reinforced soil were also prepared, and the

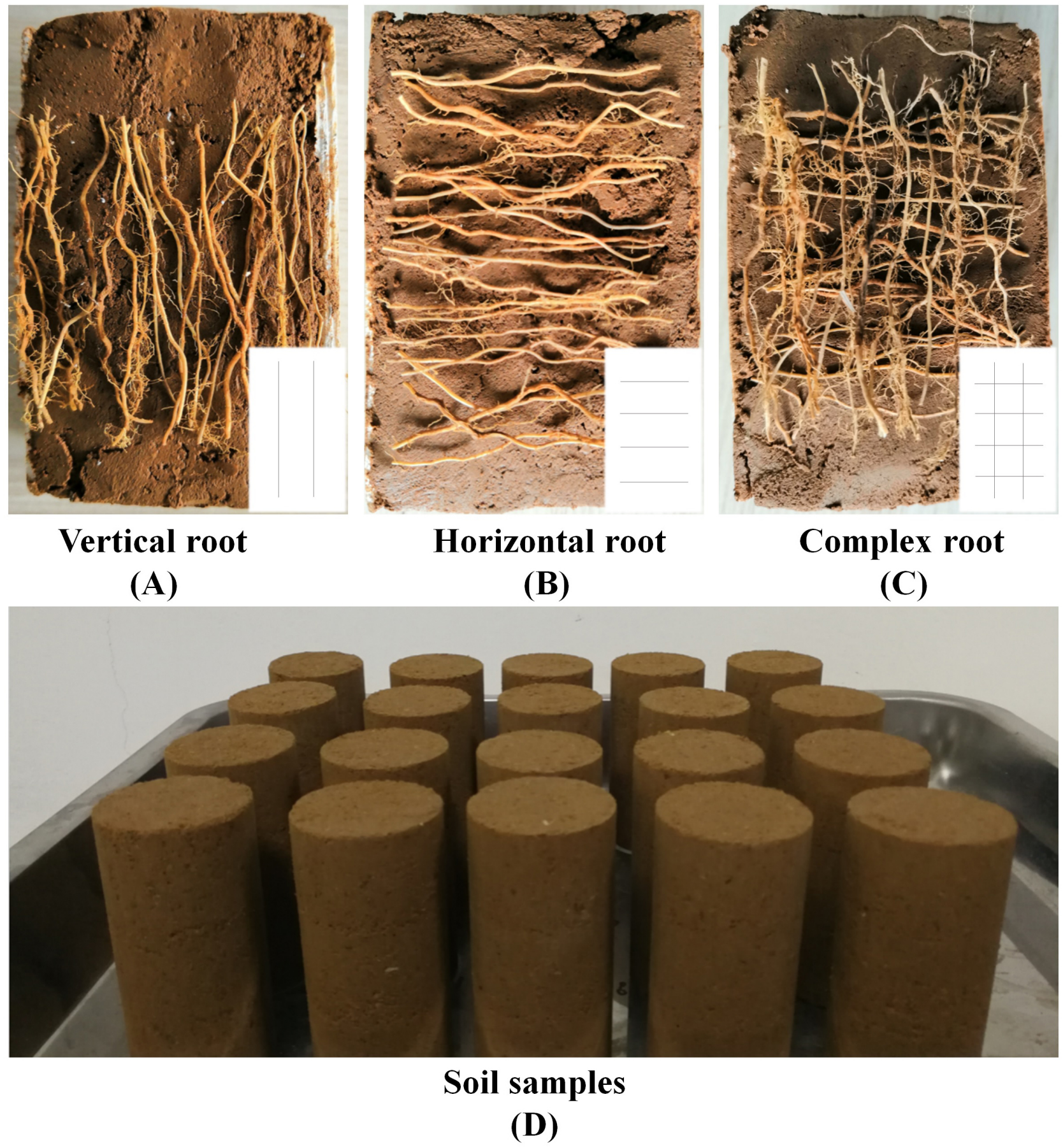

**Figure 2 Root distribution patterns in the triaxial test.** (A) Vertical root. (B) Horizontal root. (C) Complex root. (D) Soil samples.

preparation processes for the reinforced and un-reinforced samples were consistent except that no roots were present in the un-reinforced specimens.

A prepared specimen was put into the pressure room on which 20 kPa of confining pressure was applied. Water entered the specimen base until it flowed from the upper surface, and the constant head was controlled at 1.2 m. The saturated specimens were obtained when the inflow water and the overflow water were equal.

Therefore, the un-reinforced and reinforced samples with root content (0.25%, 0.50%, and 0.75%), root distribution pattern (HR, VR, and CR), confining pressure (50, 100, and 150 kPa), were performed at the different conditions of shearing rate (0.012 mm $min^{-1}$ for CD, 0.12 mm $min^{-1}$ for CU, and 0.9 mm $min^{-1}$ for UU). The shear strengths of the soil-root composites and un-reinforced soil specimens were measured by triaxial testing with 15% of the axial strain (*Zhang et al., 2010*). All conditions were repeated three times.

### Generalized equivalent confining pressure (GECP)

The GECP was derived from the traditional equivalent confining pressure. *Gray & Al-Refeai (1986)* analysed the failure mechanism of reinforced sandy soil via a triaxial test and derived the expression of traditional equivalent confining pressure under drained conditions (*Moroto, 1992*; *Li et al., 2017*) :

$$\Delta\sigma_{3t} = \sigma_3 \frac{\Delta\sigma_{1f}}{\sigma_{1f}} \tag{1}$$

where $\sigma_3$ and $\Delta\sigma_{3t}$ represent the confining pressure and traditional equivalent confining pressure, respectively, and $\Delta\sigma_{1f}$ represents the deviator of the failure principal stresses of reinforced and un-reinforced soil specimens under the same confining pressure of $\sigma_3$.

$$\Delta\sigma_{1f} = \sigma_{1fb} - \sigma_{1f} \tag{2}$$

where $\sigma_{1f}$ is the failure principal stress of un-reinforced soil under a confining pressure of $\sigma_3$ and $\sigma_{1fb}$ is the failure principal stress of reinforced soil under a confining pressure of $\sigma_3$.

The expression of traditional equivalent confining pressure is proposed for sandy soil under drained conditions, in which cohesion has not been considered (the cohesion of the sandy soil is 0). Meanwhile, the function of plant roots in reinforced soil is evaluated, which does not take the effect of the drained condition into account.

To avoid these limitations in the traditional equivalent confining pressure, *Huang et al. (2007)* proposed the GECP of cohesive soil and soil-root composites under different drainage conditions, considering that the Mohr–Coulomb strength theory is also obeyed in reinforced soil. GECP is the difference in confining pressure between the reinforced and un-reinforced soil specimens at the same shear strength (Fig. 3). The equation of GECP was derived based on cohesive soil and un-drained condition were comprehensively considered, the traditional equivalent confining pressure of sandy soil can also be realized in the equation of GECP when sandy soil be regarded as a special cohesive soil which with the cohesion is 0. Therefore, to distinguish the expression of traditional equivalent confining pressure, the GECP is expressed as $\Delta\sigma_{3g}$.

The limited balance equation of un-reinforced soil is as follows:

$$\sigma_{1f} = \sigma_3 K_p + 2\mathrm{c}\sqrt{K_p} \tag{3}$$

The limited balance equation of reinforced soil in terms of the difference in confining pressures between the reinforced and un-reinforced soil specimens at the same shear strength is as follows:

$$\sigma_{1fb} = \left(\sigma_3 + \Delta\sigma_{3\mathrm{g}}\right) K_p + 2c\sqrt{K_p} = \sigma_{1f} + \Delta\sigma_{3\mathrm{g}} K_p \tag{4}$$

where $\Delta\sigma_{3g}$ represents the generalized equivalent confining pressure; $K_p$ is the passive earth pressure coefficient of cohesive soil, $K_p = \tan^2\left(45^\circ + \frac{\varphi}{2}\right)$; and $c$ and $\varphi$ represent shear strength indexes.

Expression of GECP:

$$\Delta\sigma_{3\mathrm{g}} = \sigma_3 \frac{\Delta\sigma_{1f}}{\sigma_{1f} - 2c\sqrt{K_p}}. \tag{5}$$

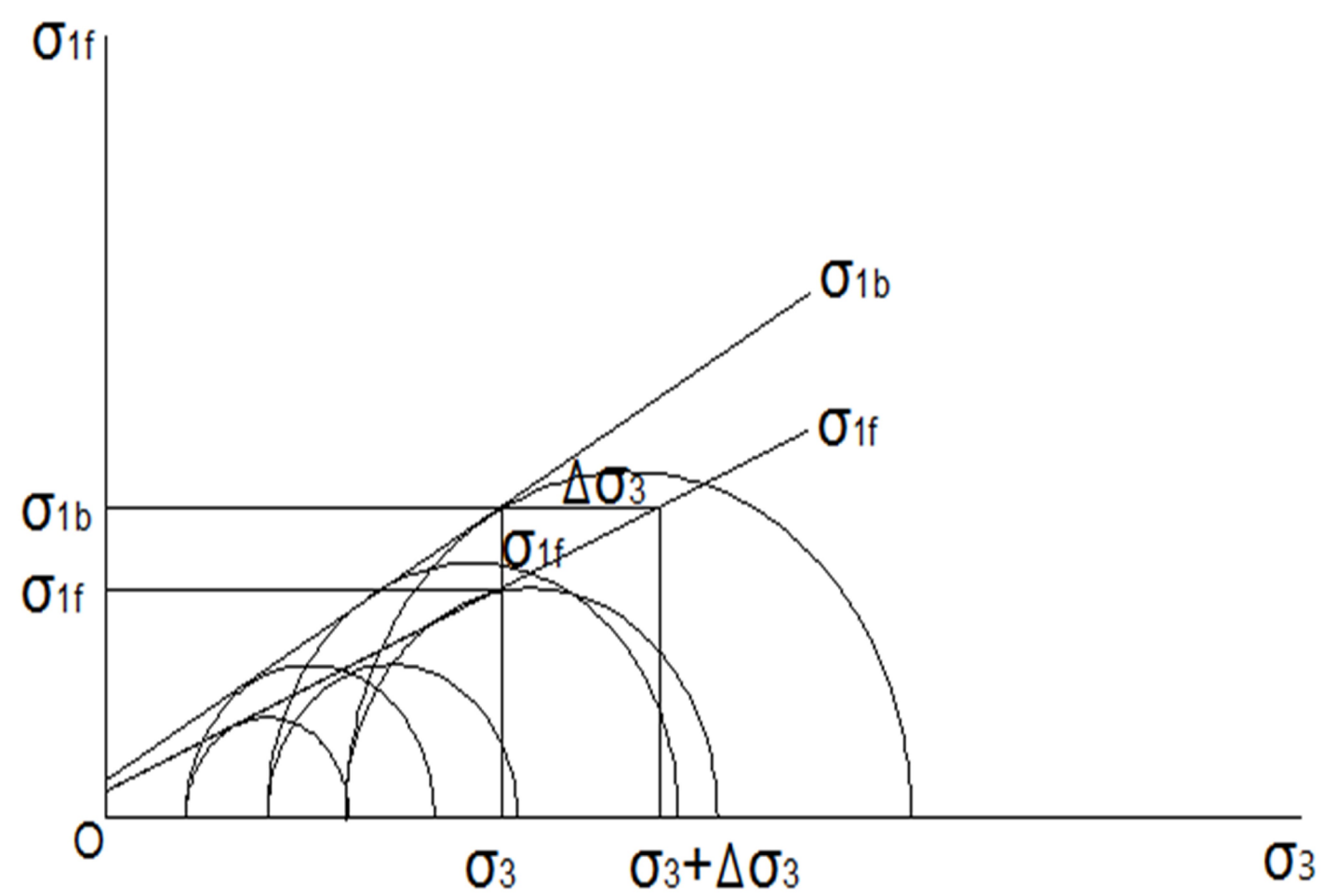

**Figure 3** **The relationship between $\sigma_1$ and $\sigma_3$ in the soil-root composite and un-reinforced soil.**

Expression (5) indicates that the GECP of the soil-root composite depends on the deviator of the failure principal stresses of the reinforced and un-reinforced soil specimens, the failure principal stress of the un-reinforced soil and the shear strength indexes of the un-reinforced soil. The expression of the traditional equivalent confining pressure is a special case when the cohesion is 0; then, expression (5). transforms into expression (1). That is, the expression of the traditional equivalent confining pressure proposed is for sandy soil, so sandy soil can be regarded as a cohesive soil when the cohesion is 0.

### Data analyses

All statistical analyses were performed using by using SPSS with version of 21.0 and Excel with version of 2010. All the data used for further analysing (*c*, $\varphi$ and *GECP*) is based on the mean value of the three replications for each condition. Therefore, no statistical test was applied when describe the results of *c*, $\varphi$ and *GECP* in this research.

## TEST RESULTS

### Shear strength indexes of soil-root composites under different drainage conditions

The shear strength indexes, *c* and $\varphi$, characterized as different trend under different drainage conditions (Table 1). For the un-reinforced soil, the *c* values were 8.24, 6.83 and 15.74 and the $\varphi$ values were 21.9° , 20.1° and 11.6° under the CD, CU and UU conditions, respectively. For the reinforced soil, the *c* values varied under different drainage conditions and root contents, while the $\varphi$ values remain basically stable under different drainage conditions due to the root content, which is different to the results of the un-reinforced soil.

Under the CD and CU conditions, the *c* values of the soil-root composites showed obviously increase. The shear strength indexes of the soil-root composites increase the most under the CD condition, with a 251.9% increase in *c* and a 45.2% increase in $\varphi$. Under the UU condition, the difference is inconspicuous in the shear strength indexes of

**Table 1** Shear strength indexes of soil-root composites.

| Control conditions | | Experimental method | | | | | |
|---|---|---|---|---|---|---|---|
| | | CD | | CU | | UU | |
| Distribution pattern | Root content (%) | $c$/kPa | $\varphi$/° | $c$/kPa | $\varphi$/° | $c$/kPa | $\varphi$/° |
| Un-reinforced soil | 0.00 | 8.24 | 21.90 | 6.83 | 20.10 | 15.74 | 11.60 |
| | 0.25 | 7.49 | 23.40 | 5.06 | 20.60 | 16.71 | 9.60 |
| HR | 0.50 | 14.10 | 25.30 | 11.41 | 21.10 | 14.03 | 11.40 |
| | 0.75 | 19.26 | 30.00 | 15.75 | 23.70 | 16.43 | 11.70 |
| | 0.25 | 14.04 | 23.40 | 12.56 | 20.10 | 15.35 | 11.20 |
| VR | 0.50 | 21.69 | 24.40 | 20.73 | 22.00 | 13.27 | 11.40 |
| | 0.75 | 27.03 | 31.60 | 22.94 | 24.90 | 11.81 | 13.50 |
| | 0.25 | 18.98 | 24.50 | 16.87 | 20.00 | 14.41 | 11.30 |
| CR | 0.50 | 23.27 | 28.40 | 22.47 | 22.90 | 14.34 | 12.00 |
| | 0.75 | 29.00 | 31.80 | 28.84 | 27.30 | 18.82 | 12.30 |

**Notes.**
CD, consolidation drained condition; CU, consolidation undrained condition; UU, unconsolidated undrained condition; HR, horizontal root; VR, vertical root; CR, complex root.

the soil-root composites when the root distribution pattern changes. The $c$ values present a complicated change trend, which mainly depends on the root distribution pattern and root content. For example, for VR soil-root composites, the $c$ decreased from 15.35 kPa to 11.81 kPa as the root content increased from 0.25% to 0.75%. However, for the HR and CR soil-root composites, the $c$ decreases and then increases with the increase in root content.

Under the same root content and drainage conditions, the $c$ values ranked in order of lower to higher were HR, VR and CR, suggesting the CR is the best at enhancing the soil shear strength.

## The GECP of *Indigofera amblyantha* roots in the reinforced soil varied by root contents

The values of GECP in the reinforced soil increased with root content (Fig. 4). Under the CD and CU conditions, when the root content is 0.75%, the GECP of the plant roots in the soil-root composites is 1.5–2.0 times that of 0.50% and more than 5 times that of 0.25%. Taking the CD condition as an example, when the soil-root composites are under 150 kPa of confining pressure and the CR content is 0.75%, the GECP of the plant roots in the soil-root composites is 106.83 kPa (Table 2). Namely, the shear strength of the soil-root composites under these conditions is equivalent to the strength of un-reinforced soil subjected to a confining pressure of 256.83 kPa. For the UU condition, the GECP of the plant roots was mainly concentrated within the range of −10 kPa to 10 kPa. The GECP of the plant roots increased as the root content increased, largely mirroring the results for the drainage condition. For the CR reinforced soil, the GECP changes from negative to positive as the root content increases, whereas the GECP is always negative under the condition of HR.

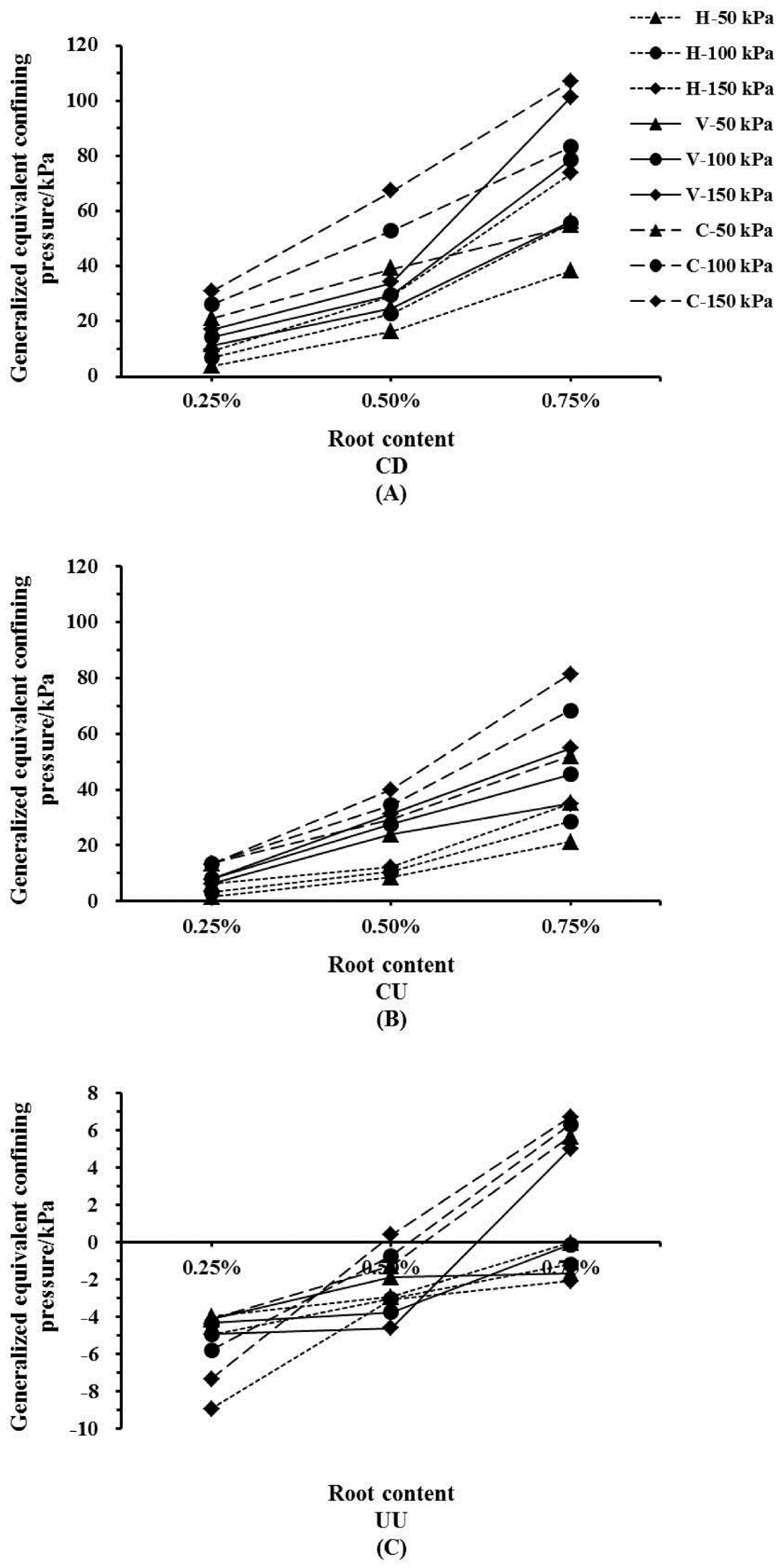

**Figure 4 The generalized equivalent confining pressure of *Indigofera amblyantha* roots in the reinforced soil varied by root contents.** (A) Consolidation drained (CD). (B) Consolidation undrained (CU). (C) Unconsolidated undrained (UU). In the legend, H, V and C denote the root distribution patterns are vertical root, horizontal root and complex root, respectively; 50, 100 and 150 kPa denote the confining pressures are 50, 100 and 150 kPa, respectively.

**Table 2** Generalized equivalent confining pressure (GECP) of *Indigofera amblyantha* roots in the reinforced soil.

| Confining pressure (kPa) | Root content (%) | CD | | | CU | | | UU | | |
|---|---|---|---|---|---|---|---|---|---|---|
| | | HR | VR | CR | HR | VR | CR | HR | VR | CR |
| | 0.25 | 3.61 | 11.23 | 21.01 | 1.61 | 6.35 | 13.72 | −3.97 | −4.10 | −4.14 |
| 50 | 0.50 | 16.13 | 24.66 | 39.01 | 8.56 | 24.03 | 29.26 | −2.90 | −1.87 | −1.26 |
| | 0.75 | 38.07 | 56.18 | 54.57 | 21.26 | 35.17 | 52.14 | −0.03 | −1.68 | 5.69 |
| | 0.25 | 6.75 | 14.06 | 26.11 | 3.36 | 8.10 | 13.65 | −4.93 | −4.32 | −5.78 |
| 100 | 0.50 | 22.68 | 29.45 | 52.80 | 10.69 | 27.71 | 34.50 | −2.99 | −3.74 | −0.70 |
| | 0.75 | 55.50 | 78.30 | 83.08 | 28.55 | 45.54 | 68.28 | −1.15 | −0.13 | 6.31 |
| | 0.25 | 9.26 | 16.87 | 30.75 | 6.30 | 7.92 | 13.29 | −8.89 | −4.89 | −7.29 |
| 150 | 0.50 | 28.92 | 33.91 | 67.05 | 12.34 | 31.30 | 40.10 | −3.07 | −4.59 | 0.41 |
| | 0.75 | 73.45 | 101.03 | 106.83 | 35.09 | 54.89 | 81.50 | −2.08 | 5.05 | 6.75 |

**Notes.**
CD, consolidation drained condition; CU, consolidation undrained condition; UU, unconsolidated undrained condition; HR, horizontal root; VR, vertical root; CR, complex root.

## The GECP of *Indigofera amblyantha* roots in the reinforced soil varied by drainage conditions and root distribution patterns

The values of GECP are positive under the CD and CU conditions, while shows from negative to positive under UU condition (Fig. 5). Generally, the values of GECP followed the sequence of CD >CU >UU. The GECP of the plant roots under CU condition increased by 5.48–74.76 when compared with those under UU condition. And the GECP of the plant roots under CD condition increased by 0.63–46.15 when compared with those under CU condition. The effect of the root distribution pattern on the GECP in reinforced soil followed the sequence of CR >VR >HR. Under the CU and CD conditions, the GECP of CR is 1-2 times that of VR and 2-5 times that of HR. The largest GECP of CR is 106.83 kPa, while it is only 21.26 kPa for HR (Table 2).

## The GECP of *Indigofera amblyantha* roots in the reinforced soil varied by confining pressure

The values of GECP increased as confining pressure increased under CD and CU conditions while showed a complex change to the confining pressure under the UU condition (Fig. 6). The GECP of the plant roots was increased when the confining pressure increased from 50 kPa to 150 kPa under CD and CU conditions. When the root content is 0.25% in the soil-root composite, GECP is negative and diminishes as the confining pressure increases under the UU condition. When the root content is 0.50%, the GECP of HR and VR is also negative, and the reduction in GECP is small compared with the results with the 0.25% root content. However, the GECP of CR changes to 0.41 kPa from −1.26 kPa due to the increase in confining pressure. For the root content of 0.75%, the GECP of the plant roots gradually increases, with the exception that the GECP of HR decreases from −0.03 kPa to −2.08 kPa (Table 2).

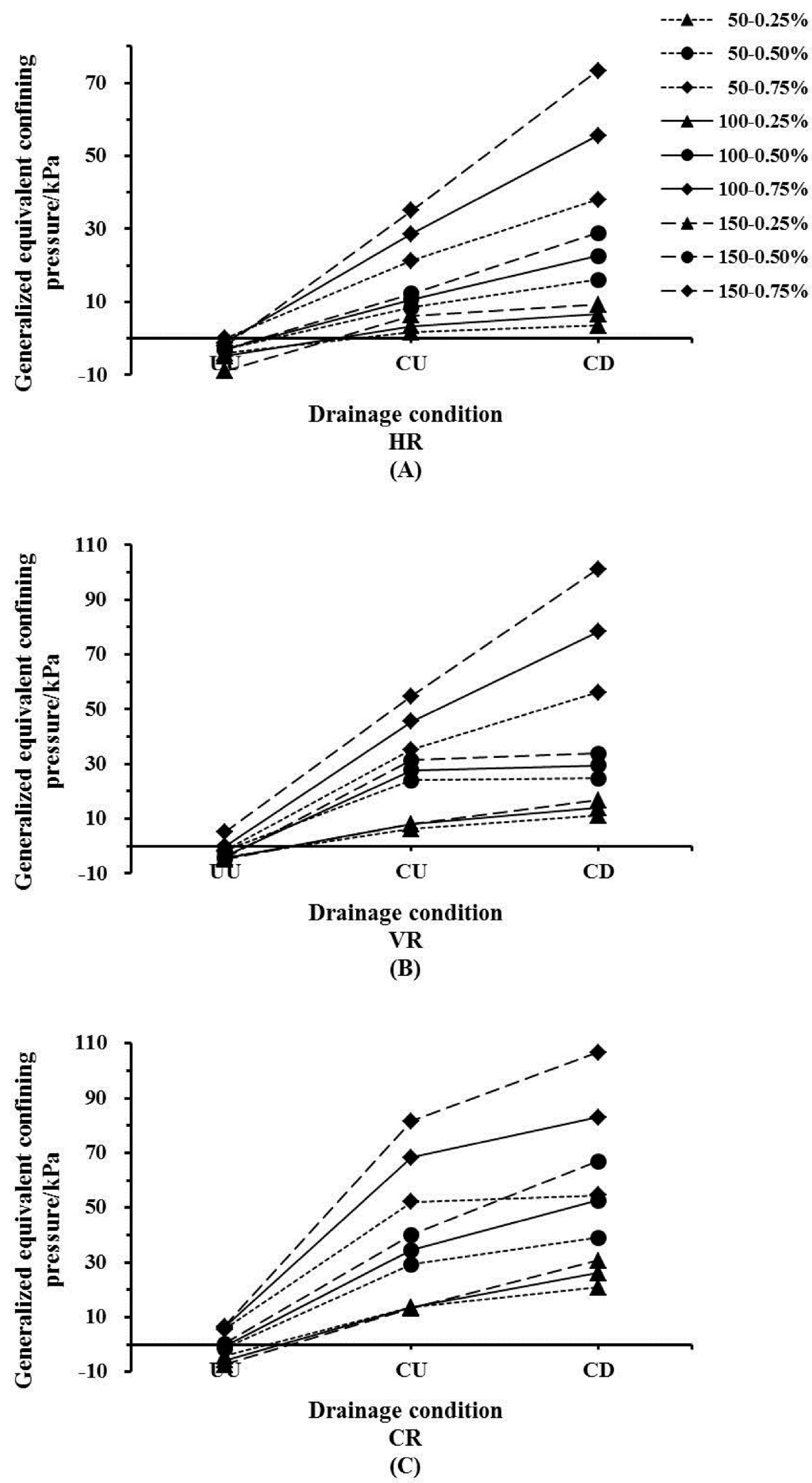

**Figure 5 The generalized equivalent confining pressure of *Indigofera amblyantha* roots in the reinforced varied by drainage conditions.** (A) Horizontal root (HR). (B) Vertical root (VR). (C) Complex root (CR). In the legend, 50, 100 and 150 kPa denote the confining pressures are 50, 100 and 150 kPa, respectively. 0.25%, 0.50% and 0.75% denote the root contents are 0.25%, 0.50% and 0.75%, respectively.

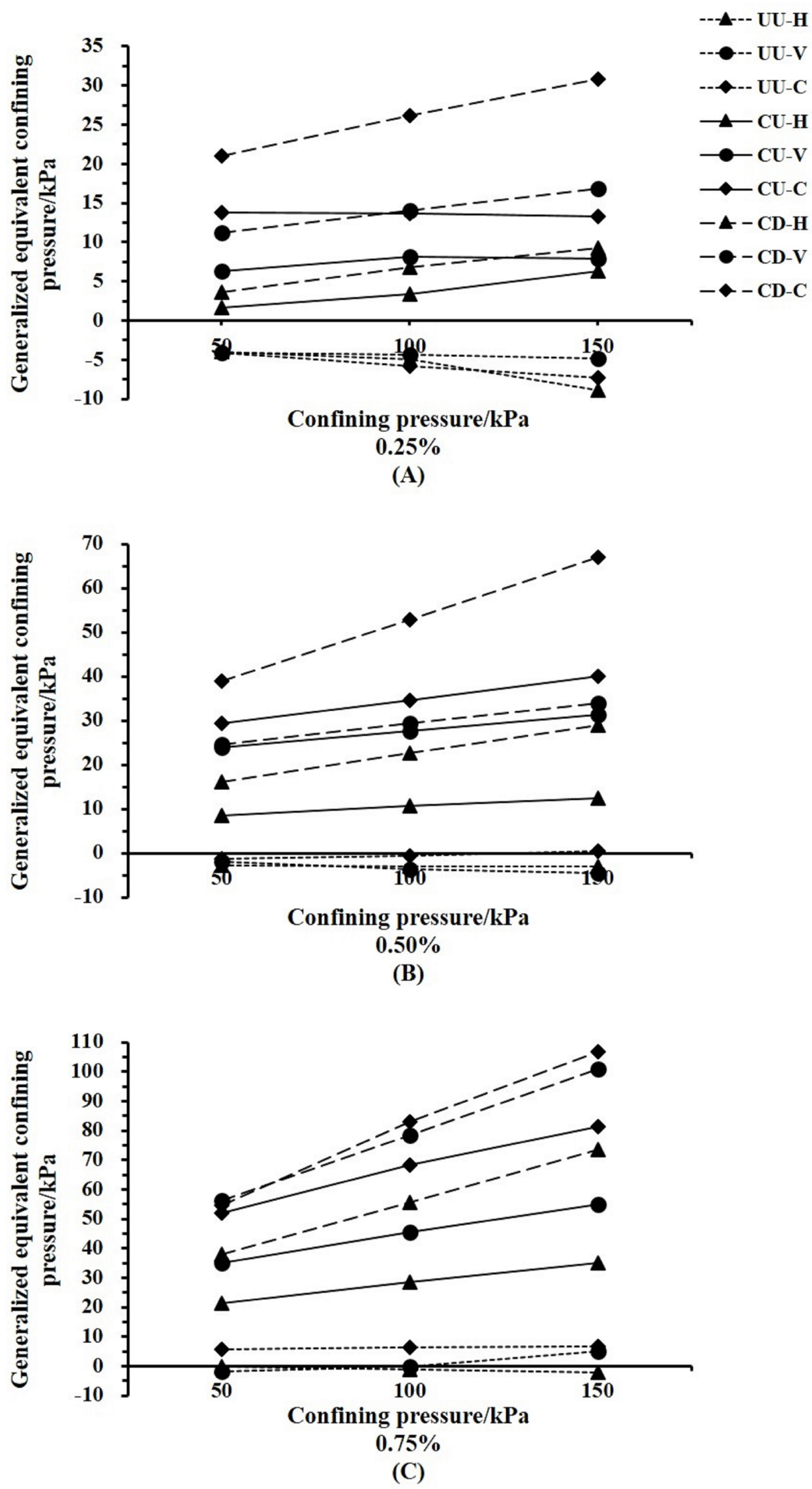

**Figure 6 The generalized equivalent confining pressure of *Indigofera amblyantha* roots in the reinforced soil varied by confining pressure.** (A) The root content is 0.25%. (B) The root content is 0.50%. (C) The root content is 0.75%. In the legend, UU, CU and CD denote the drainage conditions are unconsolidated undrained, consolidation undrained and consolidation drained, respectively. H, V and C denote the root distribution patterns are vertical root, horizontal root and complex root, respectively.

# DISCUSSION

## The evaluation mechanism based on the GECP

The soil-root composite is a composite system in which the plant roots have a high deformation modulus but the soil is weak. When soil-root composites are destroyed under an external load, dislocation occurs between soil and plant roots due to the tremendous difference in their deformation moduli. The dislocation is constrained by the frictional resistance and interlocking force between the soil particles and plant roots. Additionally, the root tensile strength and soil compressive strength are effectively equilibrated by the friction of the soil-root interface; thus, soil shear strength is improved (*Waldron, 1977*; *Waldron & Dakessian, 1981*; *Wu et al., 1988*; *Wu & Watson, 1998*; *Fan & Su, 2008*).

The reinforcing effect of plant roots on soil is mainly manifested by the addition of cohesion (*Ali & Osman, 2007*; *Normaniza, Faisal & Barakbah, 2008*), and the internal friction angle is mainly related to the soil particle structure (*De Baets & Poesen, 2006*). The phenomenon that plant roots affect the cohesion rather than the internal friction angle of the soil-root composites can be explained by the fact that living plant roots are flexible (*Huang et al., 2007*). In addition, the root content to soil mass ratio is small in the soil-root composite, although as the root content increases, the soil structure does not greatly change, so the variation in the $\varphi$ value is small (*Chegenizadeh & Nikraz, 2012*).

Compared with the Wu-Waldron model, the evaluation mechanism based on the GECP possesses the following merits: (1) different drainage conditions can be considered; (2) different stress–strain characteristics of the cohesive soil and sandy soil can be simulated; (3) the effect of drainage condition, root content and root morphology on the reinforced soil can be intuitively mirrored by the GECP, accurately and reliably. There are some possible points to discuss: for instance, Ingold (1983) showed that the shear strength of soil-root composites is worse under undrained conditions than under other conditions, but the results from our specimens do not agree.

## Effect of the root characteristics in the reinforced soil

The $c$ values of the soil-root composites showed obviously increase under the CD and CU conditions, while it presents a complicated change trend under UU condition (Table 1). The contact area increases gradually with the increase of the root content because the plant roots can be fully in contact with the soil particles. Plant roots provide an effective lateral constraint on soil: the lateral and axial deformation of soil-root composites is reduced and the shear strength is increased compared with the results of un-reinforced soil (*Tan et al., 2019*). As an exception, the stable reinforcing effect is not clearly produced when the root content is 0.25% because the low plant root content has little effect on the shear deformation. In contrast, the bonding state of the soil is destroyed when plant roots are placed in the preparation of the soil-root composites.

However, relevant studies have shown that the shear strength of soil-root composites increases with root content until a peak value is achieved, and this peak corresponds to an optimal root content (*Tan et al., 2019*). When the root content continues to increase, the plant roots are not effectively connected with the soil particles, and plant roots come into contact. Therefore, the lateral restraint of the root system in the soil is no longer

strengthened. With root contents lower than the optimal root content, the shear strength of the soil-root composites is reduced because the relative displacement is exacerbated between plant roots. Clearly, the root content is relatively low in this paper and represents the stage of soil reinforcing. The optimal root content is not the focus of this paper, so no further discussion is provided on this topic.

Among the three root distribution types (HR, VR and CR), CR is the best at enhancing the soil shear strength (Table 2).HR does not effectively reinforce soil when the root content is low because the soil integrity is destroyed and there is a smaller contact area between soil particles and the root system. However, when the root system is decussately placed in specimens, the root system bears some of the horizontal shear force and limits soil lateral deformation because of the interaction between the soil particles and the root system. Meanwhile, the rigid modulus of the soil-root composites is notably improved, which is mainly reflected in the increase in the compression modulus of the specimens, and the soil deformation is effectively restrained (*Lewis, 1956*).

## Effect of the different drainage conditions in the reinforced soil

Generally, specimens are consolidated to obtain different void ratios and left undrained to keep the void ratio constant (*Mun et al., 2016*). For soil-root composites, the initial porosity of specimens is small under consolidated conditions, and the concave-convex structure of the root surface is in contact with some soil particles. When specimens are loaded, more energy is required to overcome the interlocking force between the soil particles and plant roots. Therefore, the curve describing the relationship between the large principal stress difference and the axial strain in the soil-root composites is steeper than that for the unconsolidated specimens (*Cazzuffi & Crippa, 2005*).

In the UU triaxial test, the soil moisture content and initial porosity are high in the specimens. On the one hand, the decrease in electrolyte concentration greatly thickened the water film around the soil particles, which increased the space of the soil-root interface. Furthermore, the effective surface area of the root-soil interface decreases so that the interlocking force of the soil particles on the root system is reduced. On the other hand, the lubrication effect of water reduces the friction between soil particles and the root system, and then a soft sliding surface is formed at the soil-root interface (*Fan & Su, 2008*). In addition, the confining pressure applied to the specimens is offset by the pore water pressure based on the assumption that the volume of the specimens does not vary. The effective stresses of the specimens remain stable, so the strength envelope is relatively flat, therefore, the value of $\varphi$ is not obviously changed (*Operstein & Frydman, 2000*).

## Effect of the different confining pressure in the reinforced soil

The values of GECP in the reinforced soil increased with root content (Fig. 4). The density of the soil-root composites increased as the confining pressure increased, resulting in an increase in the soil quality per unit volume and a reduce in the soil particle gap. And the plant roots could interlock with soil particles more tightness because of the reducing in the soil particle gap, which limits the lateral deformation of soil. On the other hand, an increase in the specimen density increases the number of soil particles in contact with

the root surfaces, resulting in a larger contact area and presumably a higher cohesion in soil-root composites (*De Baets et al., 2008*; *Abernethy & Rutherfurd, 2010*).

The values of GECP are positive under the CD and CU conditions, while shows from negative to positive under UU condition (Fig. 5). The phenomenon that the GECP varies from a negative value to a positive value occurs as the root content increases (Figs. 4 and 5). This phenomenon can be explained by the fact that fewer plant roots enhance the water transport and the lubrication of the soil-root interfaces. The soil shear strength is mainly borne by soil skeleton, which is formed by the free arranging and binding of soil particles, and the biting force and bonding force between soil particles are sensitive to water when the root content is low. However large porosity exists between the interfaces of soil-root, high confining pressure accelerates soil particles gap is filled and the organic calcium of soil particles is dissolved, and the deformation resistance of soil skeleton is reduced (*Pierret et al., 2007*). Thus, fewer plant roots enhance the water transport and the lubrication of the soil-root interfaces. Inversely, high root content can limit soil lateral deformation and effectively reduce soil compression deformation, which is conducive to the dissipation of excess pore water pressure, which delayed change of pore water pressure and increased soil effective stress. However, the reinforcing effect of root distribution patterns based on different confining pressures has yet to be studied.

The GECP of the plant roots decreased and increased as confining pressure increased under smaller and higher root contents, respectively (Fig. 6). When specimens are subjected to high confining pressures, the soil particles become highly compacted. A smaller number of plant roots placed in the specimens has little influence on the density of the soil particles and the soil-root contact area. Therefore, the greater the confining pressure is, the smaller the reinforcing effect of plant roots in reinforced soil. However, for higher root contents, a high confining pressure will make the redundant plant roots fully contact the soil particles in the specimens. The soil particles at the root-soil interface will rearrange until the reinforcing effect of the plant roots is effectively exerted. Therefore, the contribution of plant roots to soil strength under a high confining pressure is greater than that under a low confining pressure.

## CONCLUSION

An evaluation based on the GECP is applied to assess the reinforcing effect of *Indigofera amblyantha* roots on soil. The results reflect that the main function of plant roots in reinforced soil is to change the soil cohesion but not to change the internal friction angle under different drainage conditions. The c values of the soil-root composites showed obviously increase under the CD and CU conditions, while it presents a complicated change trend under UU condition. Under the CD and CU conditions, the c values of the soil-root composites showed obviously increase. The reinforcing effect of the root content in reinforced soil followed the sequence of 0.75% > 0.50% > 0.25%, and the $c$ values ranked in order of lower to higher were HR, VR and CR.

The values of GECP in the reinforced soil increased with root content, and it is positive under the CD and CU conditions, while shows from negative to positive under UU

condition with an sequence of CD > CU > UU. The values of GECP increased as confining pressure increased under CD and CU conditions while showed a complex change to the confining pressure under the UU condition. Therefore, the GECP can be used as an intuitive and credible indicator to quantitatively evaluate the reinforcing effect of plant roots on soil and helps to explain the soil reinforcement mechanism of plant roots.

The results in this research are based on experiments on one soil type with the *Indigofera amblyantha* root. However, the soil characteristics (e.g., texture) and the root of different plant species have great influence on the shear strength of reinforced soil, which of course affect the influence of root contents on the shear strength of reinforced soil. Therefore, more researches under different soil characteristics with different root should be investigated in the future to verify the results obtained in this research.

### Funding

This study was financially supported by the National Key Research & Development Plan of China (2017YFC0504902), Research Fund for Excellent Dissertation of China Three Gorges University (2020BSPY003), the National Natural Science Foundation of China (41807068), the CRSRI Open Research Program (CKWV2018488/KY), and the open fund of Key Laboratory of Geological Hazards on Three Gorges Reservoir Area, Ministry of Education (China Three Gorges University) (2018KDZ06). The funders had no role in study design, data collection and analysis, decision to publish, or preparation of the manuscript.

### Grant Disclosures

The following grant information was disclosed by the authors:
National Key Research & Development Plan of China: 2017YFC0504902.
esearch Fund for Excellent Dissertation of China Three Gorges University: 2020BSPY003.
National Natural Science Foundation of China: 41807068.
CRSRI Open Research Program: CKWV2018488/KY.
Key Laboratory of Geological Hazards on Three Gorges Reservoir Area, Ministry of Education (China Three Gorges University): 2018KDZ06.

### Competing Interests

The authors declare there are no competing interests.

### Author Contributions

- Ping Guo and Lun Zhang performed the experiments, analyzed the data, prepared figures and/or tables, and approved the final draft.
- Zhenyao Xia and Wennian Xu conceived and designed the experiments, authored or reviewed drafts of the paper, and approved the final draft.
- Qi Liu conceived and designed the experiments, performed the experiments, analyzed the data, authored or reviewed drafts of the paper, and approved the final draft.

- Hai Xiao conceived and designed the experiments, analyzed the data, authored or reviewed drafts of the paper, and approved the final draft.
- Feng Gao performed the experiments, prepared figures and/or tables, and approved the final draft.
- Mingyi Li and Yueshu Yang analyzed the data, prepared figures and/or tables, and approved the final draft.

### Data Availability

The raw measurements are available as a Supplementary File.

### Supplemental Information

Supplemental information for this article can be found online at http://dx.doi.org/10.7717/peerj.10064#supplemental-information.

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
