# Peer review of "The mechanism of the plant roots’ soil-reinforcement based on generalized equivalent confining pressure"

_PeerJ, doi:10.7717/peerj.10064_

## Round 0.1 · original submission · Major Revisions

English language must be improved before peer-review can start on your manuscript.

---

## Round 0.2 · Major Revisions

This manuscript need to be deeply improved. Some important information is missing in the Introduction. Similarly, the methodology has to be better detailed. Furthermore, the importance of this study should be more clear indicated in the manuscript.

Reviewer 1 ·

Basic reporting

The authors present an experimental investigation of the mechanical effect of a root system on the shear strength of a particular type of soil. They vary root content, root placement (vertical, horizontal, mixed), confining pressure and testing conditions. The paper is overall suitable for publication in PeerJ with significant results. The presentation is globally correct, sometimes a bit rough and difficult to understand. It would have been good to provide at least one image of the samples. The results are given in terms of a Table and several plots, and discussed in full detail later. The concept of generalized effective pressure is used to quantify the effects of roots. I understand this by stating that part of the confining load is transferred to the root system. The "geometrical cohesion" induced by the roots, provides an equivalent description.

Experimental design

The experimental design is based on classical testing. The sample preparation is well-explained.

Validity of the findings

The discussion of the results is well conducted. I believe it would have been more elegant to discuss the results as they are presented. But this is a question of style. I think this work provides new insights about the effect of roots on the shear strength of soils and merits publication.

Additional comments

I have a number of suggestions below, which the authors may use to improve the paper.

lines 18-19 : I do not understand this sentence : …the generalized equivalent confining pressure (GECP), which was equivalent to confining pressure,…

The abstract should provide a clear statement about the content without referring to the definitions of terms like GECP. This term is also used in lines 34-36 to conclude that this is a convenient quantity while the reader has no idea of what it is. Please define it in the abstract. 

Lines 58-59 : What is c ? I do not understand this sentence.

Lines 65-67: Please explain what are the numbers 0.34-0.50. The sentence is not clear.

Line 70: « Shallow landslides are highly consistent with rainfall » What does it mean? 

Lines 159-160: The sentence is quite unclear.

Line 162: Where does this equation come from? Why square-root of K_p?

Line 171: Equation (5) is simply the same as equation (2) in which a cohesive strength (equal to 2c sqrt(K_p)) has been added to the failure axial stress.

Figure 4: Please give more details in the caption of the figure. What does 50-0.25% means in the figure. The quality of the figures are quite low.

Line 255: Please provide a brief presentation of the Wu-Waldron model.

Line 312: What is meant by « the value phi is trivial »?

Lines 325-326: What is meant by « A decrease in the soil particle gap is more conducive to making plant roots interlock with soil particles »?

Lines 332-334: I do not understand why fewer plant roots enhance the water transport and the lubrication of the soil-root. I would expect exactly the inverse.

Line 346: « evaluate » is repeated two times.

Reviewer 2 ·

Basic reporting

Line 62, missing some background sentences explaining the Wu-Waldron model, is this a standard model in this field?

Line 69, I found the problem statement was not very obvious for this paragraph. So if the Wu-Waldron model is not accurate, what are the missing pieces? how your study will help fill in the gap?

Line 70, "Shallow landslides are highly consistent with rainfall"? I don't understand this sentence. Do you mean shallow landslides are more common in areas receiving great precipitation? And this correlation cannot be the reason of rainfall being the major cause of slope instability. Soil characteristics (e.g., texture) can be another dominant reason.

Line 70-76, I found the importance of this paragraph was not obvious. Delete it?

Line 77, there are lots of variables can influence the soil shear strength other than these three, for example, soil texture. It was not clear that why picking these three.

Line 85, no need to have details of different levels of root contents.

Line 93, "a pH of 6.2".

Figure 1, Complex roots, I thought the ratio of horizontal to vertical was 1/2, however, this diagram was 2/1.

All font size in Figure 3-5 and Table 1-2 should be increased.

Experimental design

Line 127, I suggested authors add sentences/background to explain why choosing these three patterns of root distribution. Normally plants grow vertical roots, but I guess horizontal roots will be more common on the slope. And why the ratio of horizontal to vertical was set to be 1/2 for the Combined form?

The Method missing a section of Data analyses (methods to analyze the data, ANOVA? and in what statistical program).

Validity of the findings

Throughout the Result section, p values need to insert next to each sentence describing the significant differences. There was only one place (line 227) showing the p value, but if this was a correlation test, both p value and r value should be displayed.

Line 182, what is the "Mohr-Coulomb failure criterion"? I feel this term or analysis should be addressed in the Method section, the missing data analyses part.

Line 181, 201, 216, 226, honestly, I don't like the format of starting the first sentence (topic sentence) with Table 1 or Figure 1 showing what variables...I prefer to start a sentence summarizing the main results/effects and put the corresponding tables or figures in the parenthesis next to this sentence. The table or the figure should not be the subject, it is the main result.

Line 194, was this decrease significant? better to apply a statistical test, ANOVA or regression

Line 199, 214, 224, was there a correlation test? or just ANOVA? Because when authors used the word of "relationship" in the subheadings, readers expected to see a correlation. Or just use "GECP varied by drainage condition..", something like that.

Line 239-254, when authors discussed/explained their findings, better to put the corresponding tables/figures in the sentence, so readers know which statement was from this study and which statement was a conclusion from the literature. For example, Line 252, "the root content to soil mass ratio is small in the soil-root composite...", I believed this was a finding from this study, authors can insert (Figure X) next to it.

Line 266, confused sentence: "When root content is relatively low and increases..."?

Line 283, was this very common in the field? because plants can extend their horizontal roots to overlap with the vertical roots of their nearby plants.

Line 330, this topic sentence reads like a Result, please rewrite.

Line 354, this sentence was very similar to the sentence in line 330, please rewrite. And the following sentence is vague "Some assumptions are given to explain this phenomenon, but more research is needed to find the cause". This sentence is saying nothing! Please detail the assumption and limitations. The "Conclusion" section should include some highlights to advance this field, not just saying more research is needed...

Additional comments

This study examined the generalized equivalent confining pressure under different drainage conditions and root distribution and contents for silt loam soils rooted by Indigofera amblyantha. This manuscript missing some background information in the Introduction and data analyses in the Method. The importance of this study to the literature and the key take-home message should be more clear throughout the manuscript. Please see my comments above.

---

## Round 0.3 · Minor Revisions

Thank you for improving your manuscript according to the suggestions given. Regarding data analysis, the tests used have to be detailed properly.

Reviewer 2 ·

Basic reporting

None

Experimental design

None

Validity of the findings

None

Additional comments

First I would like to thank authors for their efforts in revising this manuscript. The structure and the problem (knowledge gap) setting in the Introduction are improved a lot. I only have one minor opinion regarding the "Data analyses" section in line 204. The current description is too simple, the performed tests (e.g., ANOVA? t-test?) should be detailed in here.

---

## Round 0.4 · accepted · Accept

I am pleased to confirm that your paper has been accepted for publication in PeerJ.

Thank you for submitting your work to this journal.